

# The impact of coronavirus disease 2019 on frozen-thawed embryo transfer outcomes

Junrong Diao[1,*], Du Aijun[1,*], Xinyan Wang[1], Shuai Zhang[1], Ying Han[1], Nan Xiao[1], Zhe Pang[2], Junfang Ma[1], Yunshan Zhang[1] and Haining Luo[1]

[1] Center for Reproductive Medicine, Tianjin Central Hospital of Obstetrics and Gynecology, Maternal Hospital of Nankai University, Tianjin Key Laboratory of Human Development and Reproductive Regulation, Tianjin, China
[2] Tianjin Medical University, Tianjin, China
[*] These authors contributed equally to this work.

Corresponding author
Haining Luo,
30317012@nankai.edu.cn

## ABSTRACT

**Background**. Coronavirus disease 2019 (COVID-19) has raised concerns about its potential effects on human fertility, particularly among individuals undergoing assisted reproductive therapy (ART). However, the impact of COVID-19 on female reproductive and assisted reproductive outcomes is unclear. In this study, we aimed to evaluate the effects of COVID-19 on pregnancy outcomes during frozen-thawed embryo transfer (FET) cycles.

**Methods**. This retrospective cohort study included 327 enrolled patients who underwent FET cycles at a single reproductive centre. The study group consisted of patients treated between 1 January 2023 and 31 March 2023 who recently recovered from COVID-19. The embryos for transfer were generated prior to COVID-19 infection. The control group consisted of patients treated between 1 January 2021 and 31 March 2021 who were not infected and did not receive a severe acute respiratory syndrome coronavirus 2 (SARS-CoV-2) vaccine. Demographic and cycle characteristics and outcomes were compared.

**Results**. A total of 160 recovered women and 167 controls were included. The primary outcome—the live birth rate—was similar between the two groups (43.8% *vs.* 43.1%, $P > 0.05$). The secondary outcomes, such as the implantation rates (41.2% *vs.* 39.3%), biochemical pregnancy rates (56.3% *vs.* 56.3%), clinical pregnancy rates (52.5% *vs.* 52.1%), early abortion rates (8.3% *vs.* 12.6%) and ongoing pregnancy rates (46.9% *vs.* 44.3%), were also similar ($P < 0.05$). According to a logistic regression model, the live birth rate did not decrease after SARS-CoV-2 infection after adjusting for confounding factors (adjusted OR (95% CI) = 0.953 (0.597∼1.523)). Regardless of stratification by age or the number of embryos transferred, the differences remained nonsignificant. Subgroup logistic regression demonstrated that the time interval from infection to transplant had no significant influence on the live birth rate.

**Conclusions**. SARS-CoV-2 infection after oocyte retrieval had no detrimental effect on subsequent FET outcomes.

## INTRODUCTION

In December 2019, coronavirus disease 2019 (COVID-19) was first reported in Wuhan, China. This new type of pneumonia, caused by the highly contagious severe acute respiratory syndrome coronavirus 2 (SARS-CoV-2), rapidly spread worldwide (*Wang et al., 2020*). SARS-CoV-2 can cause multiple systemic diseases with short- or long-term effects on multiple organs (*Caramaschi et al., 2021*). The pandemic has raised concerns regarding the possible effects of the virus on human fertility.

SARS-CoV-2 infects human host cells by binding the cellular receptor angiotensin-converting enzyme 2 (ACE2), and it also requires the cellular protease transmembrane protease serine (TMPRSS) (*Walls et al., 2020*; *Lukassen et al., 2020*). Most studies have shown that COVID-19 infection has a negative effect on sperm parameters and fertility since SARS-CoV-2 can cross the blood–testis barrier and be detected in male semen, but the impairment seems to resolve after one complete spermatogenic cycle (*Holtmann et al., 2020*; *Guo et al., 2021*; *Tur-Kaspa et al., 2021*; *Donders et al., 2022*). The female reproductive system also expresses ACE2 and TMPRSS2. ACE2 is expressed in all stages of follicular maturation in the human ovary, including in granulosa cells and follicular fluid (*Reis et al., 2011*), and ACE2 and TMPRSS2 are also expressed in the endometrium (*Henarejos-Castillo et al., 2020*). Owing to the low expression of ACE2 and TMPRSS2, few studies have investigated SARS-CoV-2 invasion in ovarian tissues (*Boudry et al., 2022*) and endometrial tissues (*De Miguel-Gomez et al., 2022a*). *Orvieto, Segev-Zahav & Aizer (2021)* reported that couples with COVID-19 had a lower proportion of high-quality embryos. *Herrero et al. (2022)* reported that patients with higher levels of IgG against SARS-CoV-2 had lower numbers of retrieved oocytes. However, some studies have reported that COVID-19 does not affect the ovarian reserve, embryonic development or pregnancy outcomes in patients who underwent postinfection *in vitro* fertilization (IVF) therapy (*Wang et al., 2021*; *Albeitawi et al., 2022*; *Youngster et al., 2022b*). Therefore, the impact of COVID-19 on female reproductive and assisted reproductive outcomes is unclear.

Frozen embryo transfer (FET) offers a chance to evaluate the independent influence of SARS-CoV-2 infection on embryo implantation. *Youngster et al. (2022a)* analysed FET cycle data and reported that patients whose oocytes were retrieved prior to the infection had decreased pregnancy rates, even in those who recovered less than 60 days prior to embryo transfer. *De Miguel-Gómez et al., (2022b)* revealed that systemic COVID-19 affects human endometrial gene expression and endometrial function. The systemic inflammation that occurs with COVID-19 (*Hu, Huang & Yin, 2021*), when sustained, may indirectly affect endometrial tissue. However, *Aizer et al. (2022)* and *Huang et al. (2023)* reported that COVID-19 infection or vaccination did not have measurable detrimental effects on implantation in subsequent FET cycles. Therefore, the associations of COVID-19 with FET outcomes are unclear.

At the end of 2022, the Chinese government changed its pandemic prevention and control policies,and immediately, COVID-19, which was dominated by the Omicron variant, spread nationwide. In this context, to neutralize any potential viral effects on

sperm parameters, the ovarian reserve and embryonic development, we evaluated the impact of COVID-19 on pregnancy outcomes in FET cycles.

## METHODS

### Study design and participants

This was a single-centre retrospective cohort study. Data in this study were retrieved from the electronic medical records system of the Center for Reproductive Medicine at Tianjin Central Hospital of Obstetrics and Gynaecology. Prior to IVF treatment, all infertile couples underwent a systematic infertility-related assessment, including collection of the couple's medical history, assessment of the female serum endocrine level, ultrasound examination, fallopian tube patency examination, hysteroscopy if necessary, and routine semen analysis. All of the data were recorded in the electronic medical records system. The study group consisted of patients who underwent FET cycles between 1 January 2023 and 31 March 2023. Oocyte retrieval for all patients was performed prior to SARS-CoV-2 infection. On the day of the FET, each patient volunteered to complete a questionnaire about COVID-19 while waiting for embryo transfer in the waiting area. The questionnaire included questions concerning whether the patient had been infected with SARS-CoV-2, how the infection was confirmed, the time of infection, the symptoms at the time of infection, the duration of symptoms, and whether the menstrual pattern changed in terms of the cycle length and bleeding volume after infection (details in Supplemental Information). According to the questionnaire information provided by the participants, some women were confirmed to be infected with COVID-19 by a real-time PCR assay conducted *via* nucleic acid testing of nasopharyngeal swabs from medical institutions, and some were confirmed *via* a positive self-administered rapid SARS-CoV-2 antigen test kit. However, in the context of the nationwide outbreak of the epidemic and the serious shortage of medical resources, some women did not undergo PCR or antigen testing to confirm COVID-19 infection. To avoid the loss of this population, patients with fever and/or respiratory symptoms and an epidemiological history were classified into the symptom-based group in this study. The control group consisted of patients with no history of past infection who underwent IVF treatments with FET during the same period and those who were treated between 1 January 2021 and 31 March 2021 (SARS-CoV-2 nondiagnosed, nonvaccinated).

The inclusion criteria for participants were (1) aged 20–42 years and (2) receiving only the first FET cycle following recovery from COVID-19. Women who fulfilled the following conditions were excluded: (1) women with a history of three or more failed FETs; (2) women with repeated spontaneous miscarriages (defined as two or more previous spontaneous pregnancy losses); (3) infertile couples with chromosomal abnormalities; (4) women with uterine factors, including uterine malformations, intrauterine adhesions, uterine submucosal myoma, uterine surgical history and other factors affecting endometrium receptivity, such as hydrosalpinx; (5) male use of a sperm donor; and (6) sequential FET cycles.

This study was conducted in accordance with the principles of the Declaration of Helsinki and approved by the ethics committee of Tianjin Central Hospital of Obstetrics

and Gynaecology (approval number: ZY2021004). Written informed consent was obtained from each participant prior to data collection.

## Endometrial preparation and embryo transfer

Endometrial preparation protocols were individually tailored by the treating physician, largely dependent on the patient's menstruation and clinical condition, and included both natural cycles and hormone replacement cycles, per usual institutional protocols. The embryos were frozen and thawed according to the instructions provided with the vitrified freezing/resuscitation solution (Japan Kato). After thawing, good-quality embryos on day 3 were defined as those with 7–10 cells, homogeneous or slightly uneven blastomere sizes, and no fragments or fragments ≤10% (*Desai et al., 2000*). The blastocysts were graded according to the Gardner and Schoolcraft quality system, and the good-quality blastocysts were those with an expansion score ≥4, an inner cell mass ≥B, and a trophectoderm score ≥B (*Gardner et al., 2000*). Embryo transfers were performed by highly experienced senior physicians. Up to two embryos were transferred per cycle. Luteal support was carried out from the day of transfer *via* dydrogesterone administered orally (20 mg/d; Duphaston, Abbott Biologicals, San Diego, CA, USA) or vaginally (90 mg/d; Crinone, Merck Serono, Switzerland) and maintained until 10 weeks of gestation, when pregnancy was established.

## Pregnancy follow-up

Serum hCG levels were initially measured 14 days after FET. Transvaginal ultrasound was performed 6∼7 weeks after FET to confirm clinical pregnancy. Subsequent pregnancy outcomes were obtained by doctors or nurses who contacted the patients by phone and recorded them in the electronic health system.

## Observation indicators

The primary outcome was live birth, which was defined as the delivery of any neonate with signs of life at 24 weeks of gestation or later (*Wang et al., 2023*). The secondary outcomes were as follows. Biochemical pregnancy was defined as an increase in the serum hCG level to greater than 25 mIU/mL 14 days after embryo transfer. The implantation rate was defined as the ratio of the total number of amniotic sacs in all patients to the total number of embryos transferred in all patients. Clinical pregnancy was defined as the presence of a gestational sac with or without a foetal heartbeat observed *via* transvaginal ultrasound. Early abortion was defined as the spontaneous termination of an intrauterine pregnancy before 12 weeks. Ongoing pregnancy was defined as an intrauterine sac with a foetal heartbeat that continued for more than 12 weeks. Preterm birth was defined as <37 weeks gestation. Low birth weight (LBW) referred to neonates born weighing less than 2,500 g. Preterm premature rupture of membranes (PPROM) was the rupture of membranes during pregnancy before 37 weeks of gestation. Gestational diabetes mellitus (GDM) was defined as a fasting plasma glucose level ≥5.1 mmol/L or a glucose tolerance test of ≥10.0 mmol/L (1 h after sugar intake) or ≥8.5 mmol/L (2 h after sugar intake). Preeclampsia was defined as having a systolic pressure ≥ 140 mmHg and/or diastolic pressure ≥ 90 mmHg after 20 weeks of gestation and proteinuria ≥ 0.3 grams in a 24-hour urine sample or random urine protein (+).

## Statistical analyses

Continuous variables are expressed as the means ± SDs and were assessed for normality. The Shapiro–Wilk test was used to evaluate the normality of the data. A normal distribution diagram (frequency histogram) was used to visually check whether the data were approximately normally distributed. Normally or approximately normally distributed data were compared using Student's $t$-test, whereas skewed data were compared *via* the Mann–Whitney U test. Categorical variables were summarized *via* counts and percentages, and Pearson's chi-square test or Fisher's exact test, when appropriate, was used to compare differences between groups. Binary logistic regression analysis was performed to determine the effects of COVID-19 on reproductive outcomes, and the data are presented as adjusted odds ratios (ORs) and 95% CIs. Variables that differed across the groups and those of clinical relevance were included as confounders in the logistic model. Possible confounders, including female age, body mass index (BMI), infertility duration, endometrial thickness, number of embryos transferred, transfer of ≥1 good-quality embryo and the luteal support protocol, were included in the logistic regression model. COVID-19 was stratified by the number of COVID-19 symptoms, duration of symptoms, days from COVID-19 infection to transfer and whether menstruation changed, and logistic regression analysis was used to further evaluate the impact of COVID-19 infection on the live birth rate. Statistical analysis was performed with SPSS (version 26.0. SPSS, Inc., Chicago, IL, USA). All tests were two-sided, and $P < 0.05$ was considered to indicate statistical significance.

# RESULTS

## Patient information

From 1 January 2023 to 31 March 2023 and 1 January 2021 to 31 March 2021, we collected 306 cycles and 270 cycles, respectively, at our IVF centre. After applying the inclusion and exclusion criteria, 160 and 167 cycles were included in the two groups. The outcomes for all participants were tracked to determine whether the patients had a live birth (Fig. 1).

The clinical characteristics of the women in the two groups are shown in Table 1. The general characteristics, including female age at oocyte pick-up (OPU) or FET, BMI, infertility duration, infertility cause, infertility type, and number of previous pregnancies and deliveries, did not significantly differ between the two groups.

For the COVID-19 group, we used three methods to determine whether women were infected with SARS-CoV-2, including nucleic acid testing (25.6%), antigen testing (41.2%), and symptom-based diagnosis (33.1%). The distribution of symptoms per woman varied from 1 to 10 when infected with SARS-CoV-2. Only one patient had 17 symptoms. We used isometric grouping to divide the number of symptoms into three grades as follows: 1∼3 (45.6%), 4∼6 (40.0%), and ≥ 7 (14.4%). The symptom duration was classified according to the natural week as follows:1∼7 days (36.2%), 7∼14 days (36.9%), 14∼21 days (10.0%), 21∼28 days (8.1%), and ≥28 days (8.8%). We also stratified the COVID-19 group by time from SARS-CoV-2 infection to embryo transfer into ≤2 months (52.5%) and >2 months (47.5%). In our study, an indicator of changes in menstruation after COVID-19 was also collected, and 31.8% of female patients reported self-perceived menstrual changes. Details

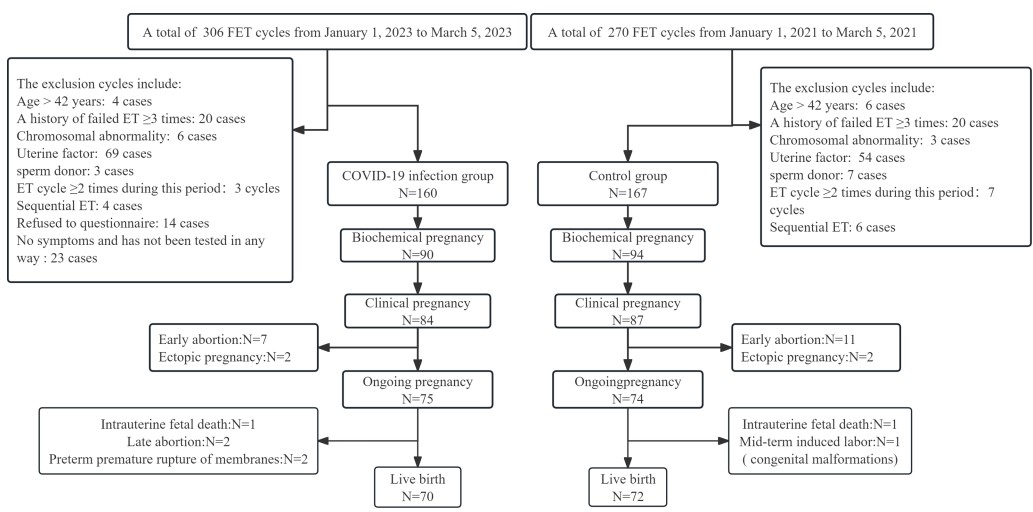

**Figure 1  Flowchart of enrolled patients and their pregnancy outcomes.**

of menstrual changes included longer or shorter menstrual cycles, increased or decreased menstrual volume, and changes in menstrual periods. Among the patients, 12.5% reported a prolonged menstrual cycle. Unfortunately, we did not collect information on how long the menstrual cycle was shortened or prolonged or whether such changes reached a level of clinical concern (Table 1).

## FET characteristics and clinical outcomes

The clinical characteristics related to FET in both groups are presented in Table 2. There were no differences in the endometrial preparation protocol, endometrial thickness, stage, or quality of the transferred embryos between the two groups. However, the number of embryos transferred, according to the proportions of single embryo transfer and double embryo transfer, significantly differed between the two groups ($P < 0.05$).

The pregnancy and perinatal outcomes of FET in both groups are presented in Table 3. There was no significant difference in the live birth rates between the two groups (43.8% *vs.* 43.1%, $P > 0.05$). Other pregnancy-related indicators, including the implantation rates (41.2% *vs.* 39.3%), biochemical pregnancy rates (56.3% *vs.* 56.3%), clinical pregnancy rates (52.5% *vs.* 52.1%), ongoing pregnancy rates (46.9% *vs.* 44.3%) and early abortion rates (8.3% *vs.* 12.6%) were also similar ($P > 0.05$). The incidences of maternal and neonatal complications such as PPROM, GDM, preeclampsia, preterm birth, and LBW were also similar between the two groups ($P > 0.05$). Binary logistic regression analysis was performed to determine the effect of a history of COVID-19 on pregnancy outcomes. Table 4 presents the results of both the crude analyses and the analyses adjusted for age, BMI, infertility duration, endometrial thickness, number of embryos transferred, transfer of ≥1 good-quality embryo and the luteal support protocol. The results revealed that the live birth rate did not decrease after SARS-CoV-2 infection after adjusting for confounding factors, and the adjusted OR and 95% CI were 0.953 (0.597~1.523). A history of SARS-CoV-2

**Table 1 Patients' baseline clinical characteristics.**

| Clinical characteristics | COVID-19 group ($n = 160$) | Non COVID-19 group ($n = 167$) | P value |
|---|---|---|---|
| Age at OPU (years) | 31.81 ± 4.046 | 32.01 ± 3.687 | 0.651 |
| Age at ET (years) | 32.11 ± 3.918 | 32.31 ± 3.716 | 0.627 |
| Body mass index (kg/m² ) | 22.41 ± 3.132 | 22.03 ± 2.485 | 0.214 |
| Infertility duration (years) | 3.89 ± 2.457 | 4.46 ± 2.710 | 0.053[*] |
| Infertility cause, n (%) | | | 0.063 |
|     Tubal factor | 90 (56.3%) | 84 (50.3%) | |
|     Ovarian factor | 25 (15.6%) | 21 (12.6%) | |
|     Male factor | 30 (18.8%) | 52 (31.1%) | |
|     Unexplained infertility | 15 (9.4%) | 10 (6.0%) | |
| Number of previous pregnancies, n (%) | | | 0.115 |
|     0 | 87 (54.4%) | 108 (64.6%) | |
|     1 | 45 (28.1%) | 32 (19.2%) | |
|     ≥2 | 28 (17.5%) | 27 (16.2) | |
| Number of previous deliveries, n (%) | | | 0.828 |
|     0 | 143 (89.4%) | 148 (88.6%) | |
|     ≥1 | 17 (10.6%) | 19 (11.4%) | |
| Number of symptoms, range | 1—17 | NA | |
| Number of COVID-19 symptoms (n) | | | |
|     1~3 | 73 (45.6%) | | |
|     4~6 | 64 (40.0%) | | |
|     ≥7 | 23 (14.4%) | | |
| Duration of the symptoms (day) | | NA | |
|     1~7 | 58 (36.2%) | | |
|     7~14 | 59 (36.9%) | | |
|     14~21 | 16 (10.0%) | | |
|     21~28 | 13 (8.1%) | | |
|     ≥28 | 14 (8.8%) | | |
| Days from COVID-19 to transfer | | NA | |
|     ≤2 months | 84 (52.5%) | | |
|     >2 months | 76 (47.5%) | | |
| Changes in menstrual pattern, n (%) | 51 (31.8%) | NA | |

**Notes.**

Continuous variables were presented as mean and (SD). Student's $t$-test or Mann–Whitney U test was performed as appropriate.

*Mann–Whitney U test.

Categorical variables were presented as n(% ). Pearson's Chi-square test was used to compare differences between groups.

$P < 0.05$ was considered statistically significant.

infection did not increase the risk of early miscarriage, and the adjusted OR and 95% CI were 0.694 (0.250~1.931).

A logistic regression model for pregnancy rates was applied to the subgroups (Fig. 2). The COVID-19 group was divided into subgroups by FET age (<35 years and ≥35 years) and the number of embryos transferred (single embryo transfer and double embryo transfer) for the binary logistic regression analysis while adjusting for confounding factors such

**Table 2 Frozen-thawed embryo transfer cycle characteristics.**

|  | COVID-19 group (*n* = 160) | Non COVID-19 group (*n* = 167) | *P value* |
|---|---|---|---|
| Endometrial preparation, n (%) |  |  | 0.638 |
|     Natural cycle | 126 (78.8%) | 135 (80.8%) |  |
|     Hormonal replacement therapy | 34 (21.3%) | 32 (19.2%) |  |
| Endometrial thickness (mm), mean (SD) | 9.03 ± 1.265 | 9.31 ± 1.519 | 0.070 |
| No. embryos transferred, n (%) |  |  | 0.000 |
|     1 | 65 (40.6%) | 34 (20.4%) |  |
|     2 | 95 (59.4%) | 133 (79.6%) |  |
| Embryo developmental stage, n (%) |  |  | 0.642 |
|     Cleavage | 154 (96.3%) | 159 (95.2%) |  |
|     Blastocyst | 6 (3.8%) | 8 (4.8%) |  |
| Transfer ≥1 good-quality embryo, n (%) | 123 (76.9%) | 129 (77.2%) | 0.937 |
| Luteal support protocols, n (%) |  |  | 0.782 |
|     1 | 58 (36.3%) | 63 (37.7%) |  |
|     2 | 102 (63.7%) | 104 (62.3%) |  |

Notes.
Continuous variables were presented as mean and (SD). Student's *t*-test was performed to compare differences between groups.
Categorical variables were presented as n (%). Pearson's Chi-square test was used to compare differences between groups.
$P < 0.05$ was considered statistically significant.

**Table 3 Pregnancy and perinatal outcomes of FET.**

|  | COVID-19 group (*n* = 160) | Non COVID-19 group (*n* = 167) | *P value* |
|---|---|---|---|
| **Primary outcomes** |  |  |  |
| Live birth, n/N (%) | 70/160 (43.8%) | 72/167 (43.1%) | 0.908 |
| Singleton | 61/70 (87.1%) | 52/72 (72.2%) | 0.027 |
| Twin | 9/70 (12.9%) | 20/72 (27.8%) |  |
| **Secondary outcomes** |  |  |  |
| Biochemical pregnancy, n/N (%) | 90/160 (56.3%) | 94/167 (56.3%) | 0.995 |
| Clinical pregnancy, n/N (%) | 84/160 (52.5%) | 87/167 (52.1%) | 0.942 |
| Embryo implantation, n/N (%) | 105/255 (41.2%) | 118/300 (39.3%) | 0.659 |
| Early abortion, n/N (%) | 7/84 (8.3%) | 11/87 (12.6%) | 0.359 |
| Ongoing pregnancy, n/N (%) | 75/160 (46.9%) | 74/167 (44.3%) | 0.642 |
| **Preterm birth**, n/N (%) | 10/70 (14.3%) | 12/72 (16.7%) | 0.695 |
| LBW in singletons, n/N (%) | 5/61 (8.2%) | 5/52 (9.6%) | 1.000[#] |
| PPROM, n/N (%) | 3/70 (4.3%) | 3/72 (4.2%) | 1.000[#] |
| GDM, n/N (%) | 4/70 (5.7%) | 4/72 (5.6%) | 1.000[#] |
| Preeclampsia, n/N (%) | 3/70 (4.3%) | 1/72 (1.4%) | 0.592[#] |

Notes.
Categorical variables were presented as n/N (%). Pearson's Chi-square test or Fisher's exact test was used as appropriate.
[#]Fisher's exact test. $P < 0.05$ was considered statistically significant.

**Table 4  Association between COVID-19 infection and clinical outcomes on crude and adjusted analysis.**

| Parameter | Crude OR (95% CI) | P value | Adjusted OR (95% CI) | P value |
|---|---|---|---|---|
| Biochemical pregnancy | | | | |
| Non COVID-19 group (n = 167) | 1 | | 1 | |
| COVID-19 group (n = 160) | 0.998 (0.645~1.546) | 0.995 | 1.140 (0.715~1.818) | 0.583 |
| Clinical pregnancy | | | | |
| Non COVID-19 group (n = 167) | 1 | | 1 | |
| COVID-19 group (n = 160) | 1.016 (0.658~1.569) | 0.942 | 1.107 (0.694~1.764) | 0.670 |
| Ongoing pregnancy | | | | |
| Non COVID-19 group (n = 167) | 1 | | 1 | |
| COVID-19 group (n = 160) | 1.109 (0.717~1.714) | 0.642 | 1.194 (0.746~1.910) | 0.461 |
| Early abortion | | | | |
| Non COVID-19 group (n = 167) | 1 | | 1 | |
| COVID-19 group (n = 160) | 0.649 (0.245~1.718) | 0.384 | 0.694 (0.250~1.931) | 0.484 |
| Live birth | | | | |
| Non COVID-19 group (n = 167) | 1 | | 1 | |
| COVID-19 group (n = 160) | 0.974 (0.629~1.509) | 0.908 | 0.953 (0.597~1.523) | 0.842 |

**Notes.**

Logistic analyses (adjusted for age, BMI, infertility duration, endometrial thickness, number of embryos transferred, transfer of ≥1 good-quality embryo and luteal support protocols).

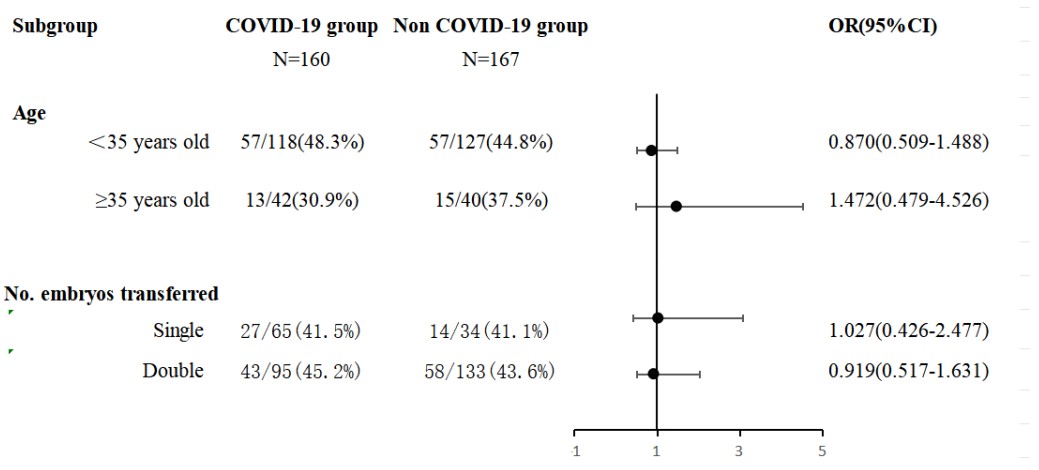

**Figure 2  Logistic regression model for live birth applied on the subgroup.** Adjusted for age, BMI, infertility duration, endometrial thickness, number of embryos transferred, transfer ≥1 good-quality embryo and luteal support protocols.

as age, BMI, infertility duration, endometrial thickness, number of embryos transferred, transfer of ≥1 good-quality embryo and the luteal support protocols were adjusted.

The results revealed that SARS-CoV-2 infection had no adverse effect on the live birth rate, regardless of whether the patients were stratified by age or the number of embryos transferred. The detailed results are shown in Fig. 2.

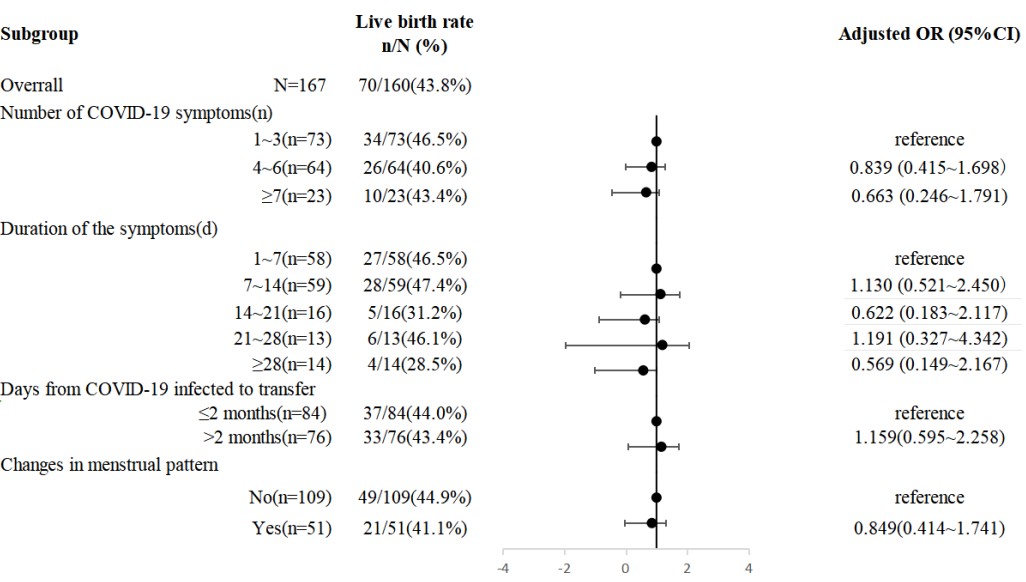

| Subgroup | | Live birth rate n/N (%) | Adjusted OR (95%CI) |
|---|---|---|---|
| Overall | N=167 | 70/160(43.8%) | |
| Number of COVID-19 symptoms(n) | | | |
| | 1~3(n=73) | 34/73(46.5%) | reference |
| | 4~6(n=64) | 26/64(40.6%) | 0.839 (0.415~1.698) |
| | ≥7(n=23) | 10/23(43.4%) | 0.663 (0.246~1.791) |
| Duration of the symptoms(d) | | | |
| | 1~7(n=58) | 27/58(46.5%) | reference |
| | 7~14(n=59) | 28/59(47.4%) | 1.130 (0.521~2.450) |
| | 14~21(n=16) | 5/16(31.2%) | 0.622 (0.183~2.117) |
| | 21~28(n=13) | 6/13(46.1%) | 1.191 (0.327~4.342) |
| | ≥28(n=14) | 4/14(28.5%) | 0.569 (0.149~2.167) |
| Days from COVID-19 infected to transfer | | | |
| | ≤2 months(n=84) | 37/84(44.0%) | reference |
| | >2 months(n=76) | 33/76(43.4%) | 1.159(0.595~2.258) |
| Changes in menstrual pattern | | | |
| | No(n=109) | 49/109(44.9%) | reference |
| | Yes(n=51) | 21/51(41.1%) | 0.849(0.414~1.741) |

**Figure 3  Subgroup analysis of live birth rate within COVID-19 group.** Adjusted for age, BMI, infertility duration, endometrial thickness, number of embryos transferred, transfer ≥1 good-quality embryo and luteal support protocols.

## Subgroup analysis of SARS-CoV-2-infected women

A stratifying analysis of the impact of COVID-19 on the live birth rate was performed. As demonstrated in Fig. 3, neither the number nor duration of symptoms at the time of COVID-19 nor the interval from infection to transplant had a significant influence on the live birth rate. A total of 14 patients (8.8%) had symptoms lasting ≥28 days. The live birth rate in this group was only 28.5% (4/14); although there was a downwards trend, which was probably affected by the sample size, the difference was not statistically significant. Changes in menstruation after infection were also not associated with the live birth rate in FET cycles (see Fig. 3 for details).

## DISCUSSION

The results of our retrospective cohort study demonstrated that prior SARS-CoV-2 infection in women had no adverse influence on subsequent FET outcomes. Moreover, different time intervals from infection to FET did not significantly affect the pregnancy outcomes.

Embryo implantation is a key step for pregnancy success, and the endometrium plays a crucial role in this process. The impact of SARS-CoV-2 on the endometrium is unclear. Some studies have suggested that the endometrium has low susceptibility to SARS-CoV-2 infection due to low ACE2 and TMPRSS2 expression, which is complicated by the limited number of endometrial samples from women with a confirmed diagnosis of COVID-19 who tested negative for SARS-CoV-2 (*Henarejos-Castillo et al., 2020*; *De Miguel-Gomez et al., 2022a*; *De Miguel-Gómez et al., 2022b*). These findings may indicate that SARS-CoV-2 does not directly invade the female reproductive tract. Instead, SARS-CoV-2 infection may

result in systemic immune dysregulation and indirectly damage some organs. Some studies have noted that inflammatory cytokine production and other biological responses caused by COVID-19, when sustained, can result in tissue damage (*Hu, Huang & Yin, 2021*; *Coperchini, Chiovato & Rotondi, 2021*). The findings of *De Miguel-Gómez et al. (2022b)* illustrated this point. They piloted a study of the endometrial transcriptomes of women with COVID-19 *via* RNA sequencing and reported that 75% of patients with COVID-19 presented with an altered endometrial gene expression profile, which manifested as the upregulation or downregulation of genes involved in immunity, inflammation, and metabolism.

FET in IVF is an excellent model for studying the impact of SARS-CoV-2 infection on implantation. Prior to our study, a limited number of studies had evaluated the impact of COVID-19 on the clinical outcomes of FET cycles. *Morris (2021)* analysed data from 143 women undergoing a single FET and reported that compared with seronegativity, seropositivity to the SARS-CoV-2 spike protein, whether from vaccination or infection, does not prevent embryo implantation or early pregnancy development. In line with this finding, *Aizer et al. (2022)* retrospectively analysed data from 428 patients undergoing 672 FET cycles and reported no significant differences in biochemical, clinical or ongoing pregnancy rates among infected, vaccinated and control patients (neither infected nor vaccinated). However, *Youngster et al. (2022a)* found the opposite opinion in their retrospective cohort study. Past infection with SARS-CoV-2 prior to FET decreased pregnancy rates, especially in women with recent infection (within 60 days), and they suggested that in cases of FET cycles with a limited number of embryos, delaying embryo transfer for at least 60 days after recovery from COVID-19 may be considered (*Youngster et al., 2022a*). Our conclusions are consistent with those of *Morris (2021)* and *Aizer et al. (2022)*. In our study, strict inclusion and exclusion criteria were adopted to eliminate the influence of uterine factors on endometrial receptivity, as well as the possible adverse effects on pregnancy outcomes of recurrent implantation failure and chromosomal abnormalities. We found that recent infection with SARS-CoV-2 had no influence on the live birth rates (43.8% *vs.* 43.1%), implantation rates (41.2% *vs.* 39.3%), biochemical pregnancy rates (56.3% *vs.* 56.3%), clinical pregnancy rates (52.5% *vs.* 52.1%) or ongoing pregnancy rates (46.9% *vs.* 44.3%) in FET cycles compared with those of unvaccinated and uninfected women in 2021 ($P > 0.05$). Moreover, it did not increase the risk of early abortion (8.2% *vs.* 12.5%, $P > 0.05$). In the logistic regression model for the pregnancy outcomes, the results remained the same, providing reassurance that past SARS-CoV-2 infection does not compromise FET treatment outcomes. Although previous studies revealed that 75% of patients with COVID-19 presented with an altered endometrial gene expression profile, our findings suggest that these changes may not persist after disease remission or may resolve in subsequent menstrual cycles because the endometrium periodically regenerates itself. However, 14 of our patients (8.8%) had symptoms lasting ≥28 days. According to the NICE guidelines (*NICE, 2020*), symptoms lasting ≥28 days are defined as long COVID-19. The live birth rate among the long COVID-19 patients was only 28.5% (4/14); although there was a decreasing trend, which was probably affected by the sample size, the difference was not statistically significant. These results offer a new possibility that deserves

our attention. A larger sample size is needed to explore the effects of the long COVID-19 on human fertility.

Existing evidence suggests that ACE2 expression increases with age, increasing the susceptibility of the endometrium in older women to SARS-CoV-2 infection (*Henarejos-Castillo et al., 2020*). This is the first study stratified by age to examine the effect of COVID-19 on FET-related clinical outcomes. In this subgroup analysis, we did not observe an adverse effect of recent COVID-19 infection on the live birth rate in older women receiving FET treatment.

In univariate analyses of FET cycles, a significant difference was observed in the number of embryos transferred. Double embryo transfer was more commonly used in the control group, whereas single embryo transfer was more commonly used in the COVID-19 group. This may be explained by the current embryo transfer strategy supporting single embryo transfer. However, in the subgroup analysis, the number of embryos transferred was not significantly associated with the live birth rate in the logistic regression model after adjusting for confounding factors.

The similarity between the SARS-CoV-2 spike protein and the human placental protein syncytin-1 has caused concern among women of reproductive age. The reason is that antibodies subsequently produced against the SARS-CoV-2 spike protein might cross-react with syncytin-1 (*Diaz et al., 2022*). Syncytin-1 plays an important role in the formation of syncytiotrophoblasts, an essential process for placenta formation (*Frendo et al., 2003*). Interference with this process might manifest as failed implantation or early pregnancy loss. Nonetheless, recent studies have noted that the similarity between the SARS-CoV-2 spike protein and syncytin-1 is limited (*Kloc et al., 2021*) and that there is no detected cross-reactivity between anti-SARS-CoV-2 spike protein antibodies and syncytin-1 (*Prasad et al., 2021*). Serum hCG levels or biochemical pregnancy (hCG > 25 mIU/mL) rates after embryo transfer can be the earliest confirmation of syncytiotrophoblast formation and embryo implantation. *Huang et al. (2023)* reported that inactivated COVID-19 vaccination had no significant effect on serum hCG levels during the earliest stage of pregnancy in women undergoing FET treatment cycles. In our study, recent COVID-19 infection did not decrease the biochemical pregnancy rate or increase the early abortion rate. Therefore, on the basis of these results, there was no cross-reactivity between anti-SARS-CoV-2 S protein antibodies and syncytin-1.

Owing to the lack of sufficient evidence, no proper recommendation can be made for an optimal time interval between COVID-19 and the postinfection FET cycle. *Youngster et al. (2022a)* suggested that in cases of FET cycles with limited numbers of embryos, delaying embryo transfer by 30–60 days after recovery from COVID-19 may be considered. In the present study, we further analysed the impact of the interval between infection and embryo transfer on pregnancy outcomes. There was no significant difference in the live birth rate between the ≤2 and >2-month intervals, suggesting a neutral impact of the intervals between COVID-19 infection and FET. Therefore, we cautiously suggest that embryo transfer may be considered after one menstrual period following recovery from COVID-19.

COVID-19 or vaccination has also been reported to cause menstruation changes, especially with respect to the menstrual length (*Barabas et al., 2022*; *Khan et al., 2022*). In our study, 31.8% of women experienced menstruation changes after COVID-19, and 12.5% experienced prolonged menstrual cycles. The reasons for these menstrual changes are complex. Menstruation is regulated by the hypothalamic–pituitary–ovarian axis, which can be affected by internal and external stimuli, including infection and changes in lifestyle. No studies have evaluated the effects of menstruation changes on FET outcomes. In the subgroup analysis, no significant effect of menstrual changes on the live birth rate was found, suggesting that these changes did not reach clinical significance. The clinical significance of these results is that physicians can give women who experienced menstrual changes an appropriate explanation.

One of the strengths of our study is the analysis of the effect of COVID-19 on the pregnancy outcomes of FET from different perspectives, including the analysis of subgroups by age and the number of embryos transferred. Moreover, this work provides clinical evidence for patients concerned about changes in menstruation. In addition, the results of this study are relatively reliable because of the strict inclusion and exclusion criteria.

The main limitation of our study is the small sample size. Another limitation is its retrospective nature, with inherent biases related to data collection and the methods of COVID-19 diagnosis, which were not standardized. Although the study adjusted for some confounding factors *via* logistic regression analysis, the results should be interpreted with caution.. Patients without any symptoms and without any test results were excluded, which may have led to the absence of data on asymptomatic infected people. In addition, the types of menstrual cycle changes were not investigated in depth.

## CONCLUSION

In conclusion, our work provides evidence that SARS-CoV-2 infection after oocyte retrieval has no detrimental effects on subsequent FET outcomes. Women can consider receiving FET after recovering from COVID-19 without having to wait 2∼3 months or more. Future larger studies are needed to validate our observations.

## ACKNOWLEDGEMENTS

The authors thank all the doctors, nurses, and embryologists in the Reproductive Medicine Center of Tianjin Central Hospital of Obstetrics and Gynecology for their help in collecting data.

### Funding
The study was funded by Tianjin Key Medical Discipline (Specialty) Construction Project (TJYXZDXK-043A). The funders had no role in study design, data collection and analysis, decision to publish, or preparation of the manuscript.

## Grant Disclosures

The following grant information was disclosed by the authors:

Tianjin Key Medical Discipline (Specialty) Construction Project: TJYXZDXK-043A.

## Competing Interests

The authors declare there are no competing interests.

## Author Contributions

- Junrong Diao conceived and designed the experiments, authored or reviewed drafts of the article, and approved the final draft.
- Du Aijun conceived and designed the experiments, authored or reviewed drafts of the article, and approved the final draft.
- Xinyan Wang analyzed the data, prepared figures and/or tables, and approved the final draft.
- Shuai Zhang analyzed the data, prepared figures and/or tables, and approved the final draft.
- Ying Han analyzed the data, prepared figures and/or tables, and approved the final draft.
- Nan Xiao performed the experiments, prepared figures and/or tables, and approved the final draft.
- Zhe Pang performed the experiments, prepared figures and/or tables, and approved the final draft.
- Junfang Ma performed the experiments, prepared figures and/or tables, and approved the final draft.
- Yunshan Zhang conceived and designed the experiments, authored or reviewed drafts of the article, and approved the final draft.
- Haining Luo conceived and designed the experiments, authored or reviewed drafts of the article, and approved the final draft.

## Human Ethics

The following information was supplied relating to ethical approvals (i.e., approving body and any reference numbers):

The study was approved by the Ethics Committee of the Tianjin Central Hospital of Obstetrics and Gynecology

## Data Availability

The raw data is available in the Supplemental Files.

## Supplemental Information

Supplemental information for this article can be found online at http://dx.doi.org/10.7717/peerj.18112#supplemental-information.

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
