# Peer review of "The impact of coronavirus disease 2019 on frozen-thawed embryo transfer outcomes"

_PeerJ, doi:10.7717/peerj.18112_

## Round 0.1 · original submission · Minor Revisions

My own minor comments include:

Decimal Place Consistency - Ensure uniformity in the number of decimal places for percentages throughout the paper and tables (e.g., "31%" vs. "12.5%").

Enhance Figure Readability - Display specific values in Figure 2 to improve the readability of results.

Increase Visuals - Convert Table 4 into a graphical format (e.g., forest plot) and include more figures to better illustrate the data.

Statistical Test Clarity - Indicate the statistical test methods used when presenting p-values in the text.

Language Improvement - Improve language clarity and grammar throughout the manuscript, potentially using professional editing services.

Data Integrity - Double-check all figures and tables for accuracy and consistency with the raw data.

Background Information - Include details about the dominant COVID-19 variants during the study period and the specific strain of the virus affecting participants.
Infection Confirmation Details - Clarify the methods used for confirming COVID-19 infections, including the exact questionnaire questions and timing of diagnoses/tests.

Outcome Justification - Provide rationale for the selection of primary and secondary outcomes, and consider including additional relevant outcomes like live birth rates and long-term follow-up.

Methodological Details - Specify the approach for testing normality of continuous variables, handling non-normal data, and selecting confounders in the logistic regression model. Provide a rationale for specific cutoffs in symptom duration and subgroup analysis.

Reviewer 1 ·

Basic reporting

1. It is suggested that the author unify the number of decimal points of the numbers in the paper, for example, "31%" in line 293 and "12.5%" in line 294 should maintain a uniform number of decimal places. The same problem exists in the table of the article.

2. It is recommended to display the specific values in Figure 2, so that the results are more readable.

3. The number of pictures in the article is too less, and it is recommended that the author convert Table 4 into picture format (Forest Plot).

4. The section of the text that provides the p-value should also indicate the statistical test method used.

Experimental design

no comment

Validity of the findings

no comment

·

Basic reporting

English Language Check:
Original: "The coronavirus disease 2019 (COVID-19) has raised concerns regarding the possible effect on human fertility, especially in individuals undergoing assisted reproductive therapy (ART).” Correction: "Coronavirus disease 2019 (COVID-19) has raised concerns about its potential effects on human fertility, particularly among individuals undergoing assisted reproductive therapy (ART)."
Original: "The second outcomes, such as implantation rates (41.2% vs. 39.3%), biochemical pregnancy rates (56.3% vs. 56.3%) and clinical pregnancy rates (52.5% vs. 52.1%) were also similar (P>0.05).” Correction: "Secondary outcomes, such as implantation rates (41.2% vs. 39.3%), biochemical pregnancy rates (56.3% vs. 56.3%), and clinical pregnancy rates (52.5% vs. 52.1%), were also similar (P>0.05)."
Original: "SARS-CoV-2 infects human host cells via binding the cellular receptor angiotensin-converting enzyme 2 (ACE2), and also needs cellular protease transmembrane protease serine (TMPRSS).” Correction: "SARS-CoV-2 infects human host cells by binding to the cellular receptor angiotensin-converting enzyme 2 (ACE2) and also requires the cellular protease transmembrane protease serine (TMPRSS)."
Original: "Most studies have shown that COVID-19 infection have a negative effect on sperm parameters and fertility.” Correction: "Most studies have shown that COVID-19 infection has a negative effect on sperm parameters and fertility.”
Original: "The embryos for transfer were generated prior to infection.” Correction: "The embryos for transfer were generated prior to the infection.”
Original: "COVID-19 was diagnosed by positive rapid SARS-CoV-2 antigen test or real-time PCR assay in nasopharyngeal swabs.” Correction: "COVID-19 was diagnosed using a positive rapid SARS-CoV-2 antigen test or a real-time PCR assay of nasopharyngeal swabs.”
Original: "Subgroup logistic regression demonstrated that the time interval from infection to transplant had no significant influence on the clinical pregnancy.” Correction: "Subgroup logistic regression demonstrated that the time interval from infection to transplant had no significant influence on clinical pregnancy.”
Original: "Embryo implantation is the key step for pregnancy success, and the endometrium is crucial for embryo implantation.” Correction: "Embryo implantation is a key step for pregnancy success, and the endometrium plays a crucial role in this process.”
Original: "Existing evidence suggests that the expression of ACE2 increases with female age. This means that the endometrium in older women may be more susceptible to SARS-CoV-2 infection.” Correction: "Existing evidence suggests that ACE2 expression increases with age, making the endometrium in older women more susceptible to SARS-CoV-2 infection.”
Introduction and Background:
The introduction gives enough background and explains why studying the impact of COVID-19 on FET outcomes is important. It references relevant literature well, giving an overview of existing studies and identifying the knowledge gap this research aims to fill.

Figures and Tables:
Based on the review, the figures and tables do not show any signs of inappropriate manipulation. They appear to accurately and consistently present the data as described in the manuscript.
Raw Data:
They are all good.

Experimental design

Research Question:
The research question is well-defined, relevant, and meaningful. It clearly addresses an identified knowledge gap regarding the impact of COVID-19 on FET outcomes.
Methodology:
The study design is rigorous, with a well-defined cohort and control group. The methods are described in sufficient detail to allow replication.
The inclusion and exclusion criteria are appropriate and clearly stated.
Ethical approval has been obtained, and informed consent from participants has been documented.

Validity of the findings

Data Analysis:
The data analysis methods are appropriate and well-executed. Logistic regression models are used to adjust for potential confounding factors, enhancing the validity of the findings.
The results are statistically sound and controlled. The ongoing pregnancy rate, early abortion rate, and other secondary outcomes are presented with adjusted odds ratios and confidence intervals.
Interpretation of Results:
The interpretation of the results is logical and aligns with the research question. The manuscript discusses the potential implications of the findings in the context of existing literature.
The conclusions are well-stated and supported by the data. The authors provide a balanced discussion of the findings, acknowledging limitations and suggesting areas for future research.

Additional comments

Strengths:
The study addresses an important and timely research question with a well-designed cohort study.
The analysis is thorough, and the results are clearly presented and discussed.
The use of logistic regression to adjust for confounders is a strength, providing more robust findings.
Weaknesses:
The sample size, while sufficient, could be larger to enhance the generalizability of the findings.
The retrospective nature of the study may introduce biases related to data collection methods.
Suggestions for Improvement:
Improve the language and grammar throughout the manuscript for better clarity and readability. Consider professional editing services if necessary.
Provide more detailed information on the methodology, especially regarding the determination of COVID-19 infection status and the stratification of subgroups.
Ensure all figures and tables are double-checked for accuracy and consistency with the raw data.

Reviewer 3 ·

Basic reporting

The study investigated the impact of COVID-19 on pregnancy outcomes in frozen-thawed embryo transfer (FET) cycles. Conducted at a single reproduction center, it involved 327 patients, comparing those who recovered from COVID-19 with a control group who neither contracted the virus nor received the vaccine. Key outcomes, including ongoing pregnancy rates and early abortion rates, showed no significant differences between the groups. The authors found that COVID-19 infection did not adversely affect FET outcomes, regardless of age, number of embryos transferred, or time from infection to embryo transfer. Overall, SARS-CoV-2 infection post-oocyte retrieval did not negatively influence FET success. Here are some suggestions to enhance the integrity of the paper:

Background: Since the infection peak in China started in December 2022 (lines 77-78), it would be helpful to include background information on whether the infections reported in your study were primarily due to a specific strain of the variant. Additionally, when discussing other findings regarding infection and embryo transfer (lines 65-76), specify the dominant variants during those periods.

Experimental design

Infection Confirmation:
• In line 89, please provide the exact questionnaire question used to assess the infection. According to lines 161-162, it seems you relied on PCR, antigen testing, and symptom-based diagnosis. Please justify the definition of symptom-based infection confirmation.
• Clarify when these COVID-19 diagnoses/tests were conducted (lines 91-92). It seems the infection history was confirmed not at the time of infection but during the completion of the questionnaire/embryo transfer.

Outcome Selection: Include a rationale for choosing the primary and secondary outcomes to contextualize their importance. Consider including other relevant outcomes such as live birth rate, maternal and neonatal complications, and long-term follow-up to provide a more comprehensive evaluation of SARS-CoV-2 infection’s impact.

Statistical Analysis:
• Include a statement on how the normality of continuous variables (lines 139-140) was tested (e.g., Shapiro-Wilk test) and specify the approach for handling non-normal data.
• Provide a rationale for the selection of confounders in the logistic regression model to clarify their relevance (lines 144-146).

Validity of the findings

Additional Details:
• Specify the timing and method of assessing infertility cause and type in the method section. It seems that infertility type was not presented in Table 1.
• The categories for the number of symptoms (1-3, 4-6, more than 7) and symptom duration (1-7 days, 7-14 days, more than 14 days) (lines 164-166) are somewhat arbitrary. Providing a rationale for these specific cutoffs would strengthen the analysis. Consider including a group for the duration of symptoms lasting ≥28 days, as used by the CDC and ONS to define long COVID, which may provide unique insights.
• Provide a rationale for the subgroup analysis of SARS-CoV-2 infected women, including how the stratification factors were chosen. Additionally, this analysis should be described in the statistical analysis section.

---

## Round 0.2 · Minor Revisions

Please refer to the two minor comments from the reviewer.

·

Basic reporting

Good

Experimental design

Good

Validity of the findings

Good

Additional comments

Language and Grammar: While the manuscript is generally well-written, a few sections could benefit from minor language polishing to ensure clarity for an international audience.
Additional Context: Providing more detailed background information on the methodologies used for diagnosing COVID-19 in the study participants could be beneficial.

---

## Round 0.3 · accepted · Accept

The authors have addressed all of the reviewers' comments